# Effects of intention understanding and brief imitative experience on the mirror neuron system: An EEG study using Japanese sign language

**Tomoki Osaki, Takehiro Minamoto**[iD]*

Faculty of Human Sciences, Shimane University, Matsue, Japan

* txminamoto@hmn.shimane-u.ac.jp

## Abstract

The mirror neuron system (MNS) is the neural system that activates when individuals both observe and execute an action. Previous studies consistently indicate that the MNS is also involved in intention understanding. In addition, MNS activity is augmented after imitating an unfamiliar action. However, it remains unclear whether conscious effort to interpret an actors' intention further activates the MNS, consequently promoting action learning. To investigate an interactive effect between intention understanding and imitative experience of unfamiliar actions, the present study measured EEG while participants observed Japanese sign language and while they imitated signing. To manipulate a level of intention understanding, we prepared two tasks: the counting task required participants to count the number of body movements in sign language, and the meaning task required participants to guess the meaning of sign language. Mu suppression, one of the indices of the MNS, was measured during three phases: pre-imitation, imitation, and post-imitation. The results revealed that the magnitude of mu suppression was comparable between the two tasks, and remained unchanged in the pre- and post-imitation phases. The null effect of the imitative experience may be attributable to cultural factors as gestural communication is less encouraged in Japanese culture, making it challenging for the MNS to learn new actions. Furthermore, absence of an effect from the task may be attributed to greater number of unfamiliar movements in sign language, resulting in the MNS preferentially processing body movements rather than the intention of actors.

## Introduction

The main function of the mirror neuron system (MNS) is action understanding [1,2]. Studies involving macaque monkeys revealed that neurons in the premotor cortex fired even when the final part of actions were concealed (e.g., grabbing an object) [3]. Similar neuronal firing was observed when monkeys only heard sounds of actions,

**Data availability statement:** All relevant data are within the paper and its Supporting Information files. The anonymized dataset of our study has been uploaded to a Zenodo repository and can be accessed with the following DOI: https://doi.org/10.5281/zenodo.17521502.

**Funding:** Japan Society for Promotions of Science #21K03130 and 24K00507.

**Competing interests:** The authors declare that the study has been conducted without any conflicts of interest.

such as paper ripping or dropping sticks [4]. These results imply that the MNS enables the understanding of the goal of an observed action or an action that is heard [1].

Intention understanding could be regarded as an extension of action understanding. For instance, greater MNS activation was observed when actions were aligned with their contexts [5]. Specifically, participants showed heightened MNS activity when observing a hand movement to pick up empty plates on a table for washing relative to when observing a hand movement to pick up a teacup from a messy table for drinking. The findings indicate that the MNS plays a critical role in comprehending what other individuals want to do, or are presently doing; namely, the intention of others [5].

Other streams of the MNS studies have shown that active experience of unfamiliar action enhances MNS activity when the person subsequently observes that action. For instance, Brunsdon et al. (2019) found that imitations of unfamiliar finger movements or tool use enhanced mu-suppression while observing the same actions subsequently [6]. In the study, the participants wore the EEG cap and observed actions presented on the monitor prior to an experimental manipulation. Then, half of the participants imitated the target actions while watching the video clips. The other half observed the video clips. The authors also manipulated repetitions of the imitation or observation. After the procedure, all the participants observed the actions again. The result showed that mu-desynchronization, an index of the MNS activity, was greater in the imitation condition than the observation condition. They also found a null effect of the repetitions, that is, mu-desynchronization was greater in the imitation condition regardless of number of repetitions. Those results indicate that brief imitation of an unfamiliar action makes the MNS more responsive when the same action was observed. It was also shown that brief imitation of an unfamiliar writing action produced greater suppression of mu-rhythm at the central channels, suggesting that activity of the MNS involves in the early stages of imitative learning [7]. In the study, participants observed unfamiliar writing movements of the Cham alphabet, which is used by an ethnic group of Southeast Asia. Under the experimental condition, participants were asked to: first, observe the writing of the Cham alphabet; second, imitate this writing; third, observe the writing again; lastly, imitate the writing again. Under the control condition, 50% of all trials were set as the "other motor experience" condition, which required participants to write two English letters after the first observation of writing the Cham alphabet, and imitate writing the Cham alphabet after the second observation. EEG recordings were acquired during the first and the second observation, and mu suppression was calculated. The results showed greater mu wave suppression under the experimental condition relative to the control condition. The two studies described above may indicate that brief motor experience of unfamiliar actions allow the MNS to embed the actions into its motor repertoire, which can be mirrored subsequently to facilitate the action observation.

Previous studies have shown that the MNS is more responsive when observed actions become familiar by contextual cues or imitative experience; however, it remains uncertain whether the effort involved in intention reading modulates MNS activity. That is, if an observer exerts more effort to understand why an individual executes an action in a particular situation (i.e., context) or to imitate the action for

a clear goal (i.e., action acquisition), the MNS becomes more responsive to the action than when the individual passively observes or merely imitates a sequence of movements. In the present study, we focused on an effect of brief action imitation on the MNS [e.g., 6, 7], and investigated whether the effort of intention reading modulates MNS activity in three phases (during the initial observation of an unfamiliar action, while imitating the action, during the second observation of the action post-imitation). By measuring MNS activity in three phases that follows the previous study [7], we were able to examine not only the effect of brief action imitation on the MNS but also the additive effect of imitative experience and intention reading on the MNS.

To achieve this goal, we used Japanese sign language as an unfamiliar action, and measured EEGs while participants observed and imitated the manual gestures used in signing. To detect MNS activation, we extracted and analyzed mu suppression while watching and imitating movement in sign language using video clips. This study tested the following hypotheses: (H1) the MNS is more activated when watching videos with a conscious effort to understand meanings of actions than when focusing only on the body movements of the actions; (H2) the MNS is more activated when watching videos of briefly imitated actions than when watching actions for the first time; (H3) there is an interaction between conscious efforts and imitation experience whereby the greatest MNS activation occurs when trying to understand the meanings of actions after imitating the actions.

To require participants to read the intention of an actor, the present study instructed participants to watch sign movements and subsequently select the meaning of the movements from three options. At the first glance, the task appears to assess inferences of action meaning (i.e., what the actor's movement meant) rather than intention understanding (i.e., why the actor made that movement). Namely, the task does not seem to capture a process to infer a goal or motivation behind certain body movements. Yet, regarding with the direct perception account of intention understanding by the MNS, perception of other's body movements has the observer immediately attribute an intentional meaning to the movements [8]. This process may hold true for gestural communication. For instance, Montgomery et al. (2007) measured activity of the mirror neuron areas (i.e., inferior parietal lobule and frontal operculum) while participants were viewing, imitating, and producing object-directed hand movements or communicative hand gestures [9]. When compared with the rest period (i.e., viewing a blank screen), the IPL and frontal operculum showed greater activation in both movements. Moreover, the activations of the IPL and the frontal operculum were quite similar between the movements. That is, the activities in the imitation and production phases were greater than the view phase in both conditions, and their time-course transition of percent signal change showed high resemblance. The result seems to indicate that gestural movement for social communication activates the MNS to grasp intentional meaning of the other actors, without target-object for body movements. In monkey, mouth movements for communication, which were not object-directed, fired mirror neurons in area F5 [10]. The result also seems to suggest that communicative body movements, including gestures, activate the MNS to read intentions that an actor tries to express in addition to superficial meaning of the gestures. Interestingly enough, when comparing the MNS-related activity between deaf signers and hearing non-signers while they were observing pantomimes or sign languages, studies have found less activity in the former groups than the latter [11,12]. Instead, deaf-signers showed greater activity in the left hemisphere language areas including the inferior frontal gyrus and superior temporal sulcus [13]. Those results indicate that deaf signers are likely to depend on semantic processing when perceiving others' manual movements. By contrast, hearing non-signers appear to count on the MNS when perceiving the movements. Collectively, we assume that hearing non-signers would show the MNS-related neural activity when they view sign-languages, which is not only for others' motion perception but also for reading their intentions behind body movements.

## Materials and methods

### Participants

Altogether, 37 undergraduate and graduate students were recruited at Shimane Univeristy (21 females; mean age = 20.61, SD = 1.71). All participants had normal or corrected-to-normal vision and normal hearing. Prior to the experiment, the study

was explained to the prospective participants and all participants provided their informed consent. The study protocol was approved by the Institutional Review Board of the Faculty of Human Sciences at Shimane University. No participants had previous experience with Japanese sign language. Participants were paid a fee of 900 Japanese yen as compensation for their participation. The recruitment period started on 13/9/2023 and ended on 28/11/2023.

## Apparatus

The program of the experiment was created using PsychoPy [ver.2023.1.3, 14]. EEG readings were conducted inside an electromagnetically shielded chamber (W2,059 mm, D2,059 mm, H2,150 mm). The chamber contained a computer monitor (23.0 inches; EV2316W, EIZO Corporation), a keyboard (Microsoft Wired Keyboard 600, Microsoft Corporation), a seat, a chin-rest (T.K.K.932, Takei Scientific Instruments Co., Ltd.), and an EEG recording system (see below for details). The computer monitor was used to display the instruction of the experiment or the stimulus to the participants. It was synchronized with another computer outside the chamber, and the experimenter could control the computer monitor inside the chamber throughout the experiment. Participants responded to the stimulus via the keyboard. The computer for recording EEGs (MacBook Pro, Apple Inc.) was placed outside the chamber and triggers were sent to this computer via a parallel port (EC1PECPS, StarTech.com) when a stimulus started or finished.

## Movies of Japanese sign language

Twenty-six videos of Japanese sign language performed by the author were prepared. Videos were recorded using the basic Camera application of iPhone XS (Apple Inc.). The length of the videos ranged from 4.30 to 6.33 seconds (mean = 5.30 s, SD = 0.60 s). Simple sign language phrases for beginners (e.g., "I have a fever since yesterday," "It's so hot today"), retrieved from Toyoda [15], were used. Each sentence consisted of three to eight words when translated into English (mean = 5.60, SD = 1.35) and was represented by three to six signs. A sample movie is seen in the S1 movie ("Long time no see, everyone. How are you?").

There were two main reasons for using sign language as an unfamiliar action. First, in the Japanese population, few people know sign language well. This allowed us to prevent participants from guessing meanings of actions when they were instructed to focus only on body movements (i.e., the counting task). Second, although signs were mostly meaningless for novices, there were some clues to guess their meanings when participants carefully paid attention to body movements to comprehend performer's intention (i.e., the meaning task). That characteristic allowed us to examine an effect of conscious effort on the MNS while observing unfamiliar actions.

Although all the videos featured different body movements, they were systematically retrieved from a single type of construct: sign language. This approach enabled us to identify the effect of conscious effort in interpreting the meaning of actions, by comparing the MNS-related EEG recording when participants attempted to understand the intended meaning of the signs performed with the recordings when participants only focus on the superficial body movements of the performer.

## EEG recording

A dense array EEG system (GES400, Magstim, Inc.) was used for EEG recording. The device amplified signals delivered from HydroCel Geodesic Sensor Net 32 channels (GES400, Magstim, Inc.), which were equipped with an EEG cap whose sockets were arranged to follow the 10–20 international systems. We prepared three different EEG caps (small, medium, and large) and assigned one of them to a given participant according to their head size. Electrodes were soaked in a solution that contained 1L of purified water, 10 mL of potassium chloride, and 5 mL of baby shampoo. Participants were requested not to use hair conditioner and/or hair products the night before the experiment, because these prevent the electrodes from attaching on the scalp.

Before wearing an EEG cap, the vertex was determined by identifying the intersection between one line connecting the nasion and inion and the other one connecting the bilateral preauricular points. The EEG cap was placed on the skull by

locating the reference electrode (i.e., Cz) on the vertex and symmetrically placing the bilateral electrodes. After wearing the EEG cap, impedance was measured throughout the channels and additional solution was added to the electrodes with impedance exceeding the standard 100 kΩ. Net Station Acquisition software (Magstim, Inc., Eden Prairie, MN) was used for EEG acquisition with a sampling rate of 250 Hz.

## Procedure

Participants were seated on the chair inside the chamber. Using the chin-rest, the visual distance from the computer monitor was fixed to 80 cm. With the EEG cap in position, two sign language tasks were administered. The counting task involved a total of 10 trials, each trial consisting of three parts (i.e., pre-imitation, imitation, and post-imitation). In the pre-imitation part, a video clip of a sign speech was played and the participants were required to count how many body movements the model made in the video. After watching the movie, the participants reported the number of movements (from 1 to 9) by pressing the number keys on the keyboard. In the imitation part, the same clip was played and the participants were instructed to imitate the body movements. The procedure in the post-imitation part was identical to that in the pre-imitation part. Therefore, participants watched the same video three times in each trial. At the beginning of each part, we presented the first frame of a video for 3 seconds to measure baseline EEG.

The meaning task was introduced after the counting task. This task also involved 10 trails each consisting of three parts. The procedure was mostly identical to that in the counting task, except that the meaning task required participants to observe a sign speech and read its meaning. In the post-imitation phase, three Japanese sentences were presented on the monitor after a video clip, and participants were instructed to select the sentence that was likely to correspond to the meaning of the sign speech. Each sentence was indexed by a digit, and participants pressed one of three keys (1–3) to answer. Among the options, one sentence represented the correct meaning of the sign speech.

All the participants performed the counting task first to prevent a risk that prior experience of the meaning task could encourage the participants to infer the meaning of sign languages while counting number of body movements. During the pre-imitation phase, post-imitation phase, and when displaying a static frame, participants were asked to place their chins on the fixing stand and refrain from moving their body or blinking as much as possible. Before both the counting task and the meaning tasks, 3 practice trials were conducted after the provision of explanations for the corresponding task. In each task, stimulus order was randomized across the participants. Fig 1 shows a flow of one trial in each task, where the first author performs sign speech in the photos. The individual pictured in Fig 1 and recorded in the supplementary movie (S1 Movie) has provided written informed consent (as outlined in PLOS consent form) to publish his image and movie.

## EEG analysis

MNE-Python [ver.1.5.1 16] was used for EEG data analysis. To assign EEG channels on the scalp surface, we selected the template Geodesics Sensor Net HydroCel 32 channels, which was embedded in MNE-Python. Fig 2 shows a spatial layout of the sensors.

In the present study, we mainly focused on mu rhythm, which is frequently used as an index of MNS activation. Mu rhythm is a range of neural oscillations with a frequency band from 8 Hz to 13 Hz, and is supposed to be recorded using electrodes placed above the sensorimotor cortex [17,18]. During rest, for example with the individual seated quietly, neurons in the sensorimotor cortex synchronously fire, increasing the power ($\mu$V2/Hz) of the mu rhythm. Conversely, movement of body parts results in weakening of the synchroneity of the sensory-motor cortices in order to perform a specific movement, decreasing the power of the mu rhythm. The desynchronization represents a shift from simple to complex activities of multiple neurons in the premotor cortex and sensory-motor cortices, which is called mu suppression [17,18]. Mu suppression is observed not only when individuals move their body parts, but also when they observe the actions of other individuals; therefore, mu suppression is considered to reflect MNS activation [17–19]. One should note the controversy regarding the distinction between mu suppression and a decrease in alpha-band oscillation, because both occupy the same frequency band (8–13 Hz) [18,20,21].

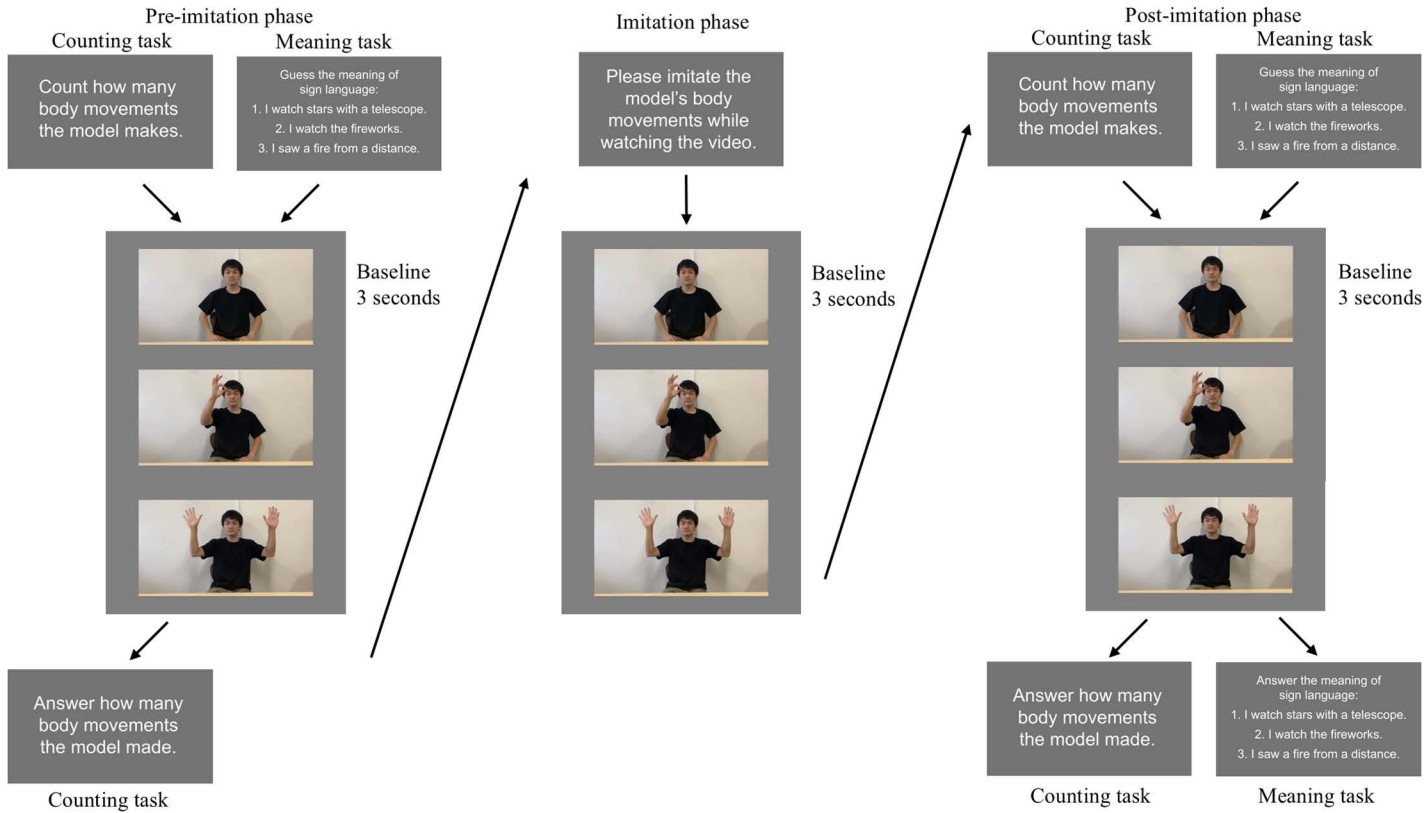

**Fig 1. A schematic diagram of the counting and meaning task.** During the pre-imitation phase (left), participants were requested to count number of hand movements the actor made in the counting task. In the meaning task, in contrast, they were instructed to guess what the sign speech meant. In the imitation phase (middle), the participants were required to imitate a sign speech, following the actor in both tasks. The post-imitation phase (right) was identical to the pre-imitation phase except that the participants needed to select one of three options for the answer in the meaning task.

Since we primarily focused on mu suppression, we analyzed EEG recordings from electrodes positioned above the sensorimotor cortex (C3, Cz, C4). Additionally, EEGs from O1, Oz, and O2 were analyzed to assess the power of alpha waves. Because mu and alpha rhythm cover the same frequency range (8–13 Hz), we can assert that the EEG data reflects mu suppression only in the presence of selective power suppression in the central sites, and not at the occipital sites, when the participants' observed and executed Japanese sign language.

Preprocessing began with the application of a bandpass filter from 1–70 Hz and a 60 Hz notch filter. Then, electroocu-lography (EOG) was computed referring to electrodes placed near the eyes (Fp1, Fp2, 18, 27, 29, 30). Independent component analysis (ICA) was performed to remove EOG-related artifacts. Subsequently, processed EEG data were epoched to obtain event-related oscillations in mu and alpha band frequency. Because each task (the counting and meaning tasks) consisted of 10 trials and each trial consisted of 3 parts (i.e., pre-imitation, imitation, and post-imitation), a total of 60 epochs were created. To make comparisons among trials easier, all the epochs had the same duration, ranging from 2.0 s prior to the beginning of a video to 4.0 s after the beginning (the shortest video was 4.30 s), although video length differed across trials. The first 1.0 s of the baseline was excluded because the EEG may be influenced by abrupt attentional shifts that occur with the onset of the stimulus. During segmentation, we excluded epochs if any of six electrodes (C3, Cz, C4, O1, Oz, O2) exceeded a peak-to-peak signal amplitude of 100 $\mu$V. Data from 5 participants were removed because all the epochs were rejected in at least one of the six conditions (2 tasks x 3 parts). In the following analysis, we included EEG data of 32 participants. Next, the time-frequency representation (TFR) was computed on the remaining epochs using the

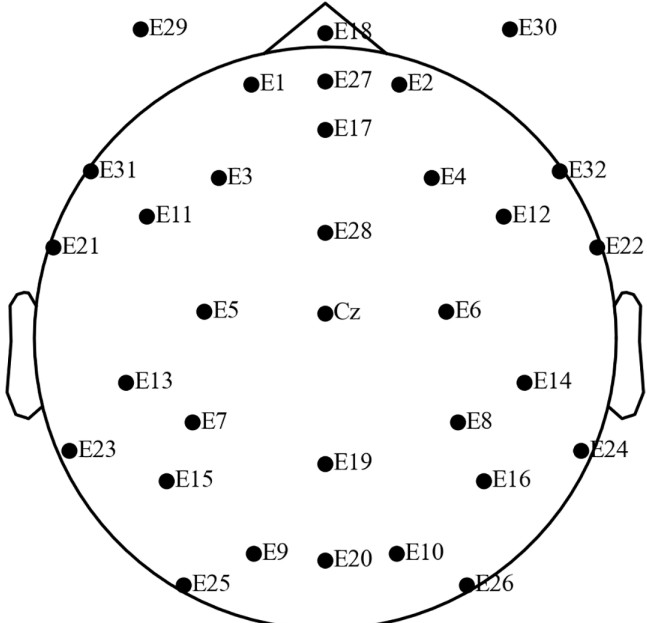

**Fig 2. Spatial layout of the EEG sensors.** Labels with an alphabet followed by a digit indicate the sensors matching the international 10-20 system. Labels with a number without alphabets indicate the sensor positions designated by the EGI Netstation registration system. The present study focused on EEG signals from the central channels (C3, Cz, and C4) and occipital ones (O1, Oz, and O2).

Morlet wavelets, and values of power were extracted. For the wavelet analysis, we used the default width value (i.e., 2 cycles) embedded in Python-MNE. Because the value is low, the obtained results appear to have better temporal resolution but may suffer from lower frequency resolution [22] due to the trade-off between temporal precision and frequency precision. The frequency range was 8–13 Hz.

Even though there have not been conclusive agreement as for an experimental method to capture baseline mu-activity, such as sitting quietly with or without visual white noise, or watching a fixation cross for several seconds, we used a static first frame of the videos, with a duration of 3.0 s, as reported by Hobson and Bishop [20]. According to the authors, the initial static frame of each movie stimulus can be used as the most appropriate baseline relative to others such as a cross fixation. Specifically, the static initial frame is less likely to produce a change of attentional state due to smooth transition to the movie, while others (e.g., cross fixation) elevate attentional level because of grater shift of visual input. The change ratio of mu power (0–4.0 s) compared to the baseline (−2.0–0 s) was calculated for each trial by initially subtracting the average baseline mu power from the time-series mu power during the tasks and dividing the subtracted value by the baseline power, based on the formula for event-related desynchronization [23]. Before calculating the change ratio of power, outliers were removed using the 2 $\sigma$ rule.

Although our primary interest was desynchronization of mu-band oscillation, that of beta-band was also analyzed because some studies reported a decrease in beta-rhythms while executing or perceiving actions [24,25]. The same wavelet analysis described above was applied except that beta-band oscillation covered the frequency range of 14–30 Hz.

### Statistical analyses

**Power analysis.** For the main EEG analysis, we used the Bayesian analysis of variance to test effects of interests (i.e., the task and imitative experience) on neural oscillation's data. It is suggested that "power" is defined from a frequentist perspective but is not directly applicable to the Bayesian framework [26]. This is because power is the long-run

probability of finding an effect, which is inherent to frequentist concept. As the Bayesian analyses employ priors provided by the precedent studies, and obtained results will be embedded into their priors for the following studies, the concept of power cannot be directly applied.

Although the present study did not use a repeated ANOVA for our analysis, we ran the power analysis (effect size = 0.25, α = .05, β = 0.8, number of measurements = 4) to compute sample size required to draw meaningful conclusion. The analysis recommended a sample size of 24, which is smaller than the size of the present study.

**Behavioral data analysis.** To confirm that participants focused on the tasks, accuracy rates were compared against the chance levels, using the one sampled t-test. Because the counting task had 9 possible responses and the meaning task had 3 possible responses, the chance level was set to 11.11% for the former and 33.33% for the latter. An α-level of 0.05 was established as the threshold for statistical significance.

**EEG data analyses.** Mu and alpha rhythm suppression were examined by comparing change ratio of powers in each phase (i.e., pre-imitation, imitation, and post-imitation) of each task (i.e., counting and meaning) against 0 (i.e., baseline), using the one sampled t-test. An alpha-level of 0.05 was established as the threshold for statistical significance. The false-discovery rate (FDR) was applied to p-values to resolve the issues associated with multiple comparisons.

We also compared mu power with alpha power in the imitation phase in each task, which aimed to examine an effect of real-movement on EEGs in central and occipital sites. The one sampled t-test was employed with an alpha-level of 0.05 as the threshold for statistical significance.

To investigate effects of the task and imitative experience, the Bayesian analysis of variance (BANOVA) was performed using R (ver.4.3.2, R Core Team, 2023) and an R package BayesFactor [ver.0.9.12–4.7 27]. We adopted the Bayesian method due to its informational advantages over conventional frequentist methods [28–31]. Specifically, the Bayesian approach presents the following strengths: first, the approach provides the full distributions of inference data, not only values [32,33]; second, it can test both alternative and null hypotheses based on the Bayes factor [34]; third, it is immune to the multiple testing problem in the frequentist approach [33].

The same analyses above were applied to beta-rhythm from the central and occipital regions.

## Distinguishment of mu and alpha rhythms

Because mu and alpha rhythms showed similar patterns in response to the experimental factors (i.e., task type and imitative experience) as seen in Results, we employed the independent component analysis (ICA) and a machine learning method to test whether two rhythms were dissociated. First, we extracted time-series oscillation data (8−13 Hz) from six channels (i.e., C3, Cz, C4, O1, Oz, O2) of individual participant in each trial, which covered the time range from −2.0 to 4.0. To simplify the following analysis, we averaged time-series data of each channel across trials and phases (i.e., pre-imitation, imitation, post-imitation). The process was performed in each task (i.e., counting and meaning task), which yielded a total of 12 time-series data in each participant (i.e., 6 channels x 2 tasks). Next, we applied ICA to 6 averaged time-series data (i.e., 6 channels) in the counting task to assign those signals to one of two independent components. The same procedure was performed in the meaning task. Then, we extracted dyadic values of contribution rate for the first and second components in each channel. Finally, we averaged the values of the central channel and those of the occipital channels in each component separately. Because each participant had averaged values from two regions (i.e., central and occipital) in two tasks, a total of 128 dyadic values were obtained (32 participants x 2 regions x 2 tasks). Fig 3 shows plots of the 128 dyadic values, where each blue dot indicates the contribution rates of the first and the second independent component belonging to the central regions. Each orange dot indicates the contribution rates belonging to the occipital regions.

To see whether the mu-rhythms were distinguishable from the alpha-rhythms, we trained the support vector machine (SVM) with the two features (i.e., contribution ratio of the first and second components) and tested whether it was able to classify the label (i.e., central or occipital regions) above chance level. The default setting was used for the SVM, where

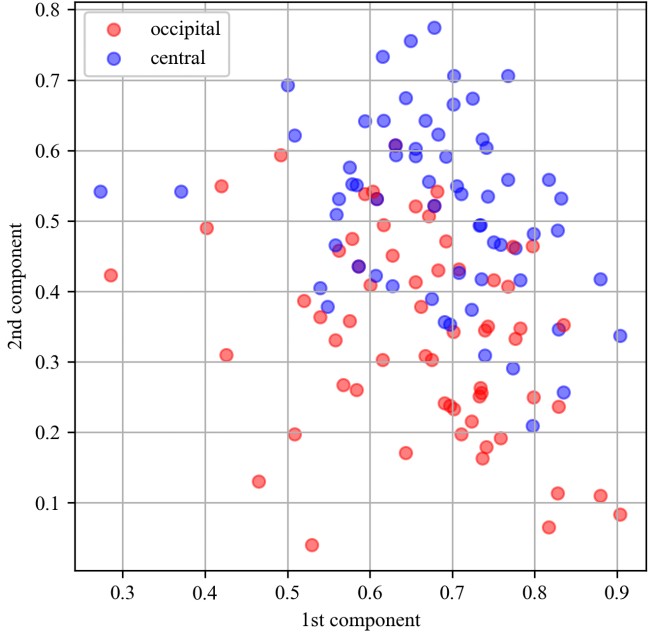

**Fig 3. Scatter plots of the contribution rate for the first and second components obtained from the independent component analysis performed on time-series data of mu- and alpha-oscillation.** Each blue dot corresponds with the contribution rates belonging to the central regions, while orange corresponds with the rates to the occipital regions.

the radial basis function was selected as the kernel and C-value for the regularization parameter was set as 1.0. The permutation test was employed to evaluate performance of the trained machine, where accuracy by the target machine was compared with that by the other machines which were trained to associate the features with randomized labels. The test was repeated 500 times and p-value was computed with the following formula: (C + 1)/ (number of permutations + 1) where C represents the number of comparisons where the randomized machine shows higher accuracy than the target machine. In addition to accuracy, we assessed the other metrics (i.e., precision, recall, and F1-score). The machine learning was performed with the scikit-learn library (https://scikit-learn.org/stable/modules/generated/sklearn.svm.SVC.html#sklearn.svm.SVC).

## Results

### Behavioral task

In the counting task, the average accuracy rate of the participants' responses was 75.95% ($SD = 14.99$). In the meaning task, we failed to collect responses in the 10th trial due to a program error. Hence, accuracy for each participant was computed by dividing correct responses by nine. The average accuracy score of all participants in the meaning task was 69.62% ($SD = 14.64$). Because both accuracy rates were significantly higher than their chance levels (counting task: 11.11%; meaning task: 33.33%), we assumed that participants sufficiently focused on the tasks (counting task: $t$ (36) = 26.31, $p < 0.001$, $d = 7.16$, 95% CI [0.71, 0.81]; meaning task: $t$ (36) = 15.01, $p < 0.001$, $d = 6.72$, 95% CI [0.65, 0.75]).

### Mu and alpha power while observing and imitating sign language

Fig 4 shows time-series mu power in each phase (i.e., pre-imitation, imitation, and post-imitation) of the meaning (1st row) and counting task (3rd row). As mentioned above, alpha power was also calculated to confirm that suppression was

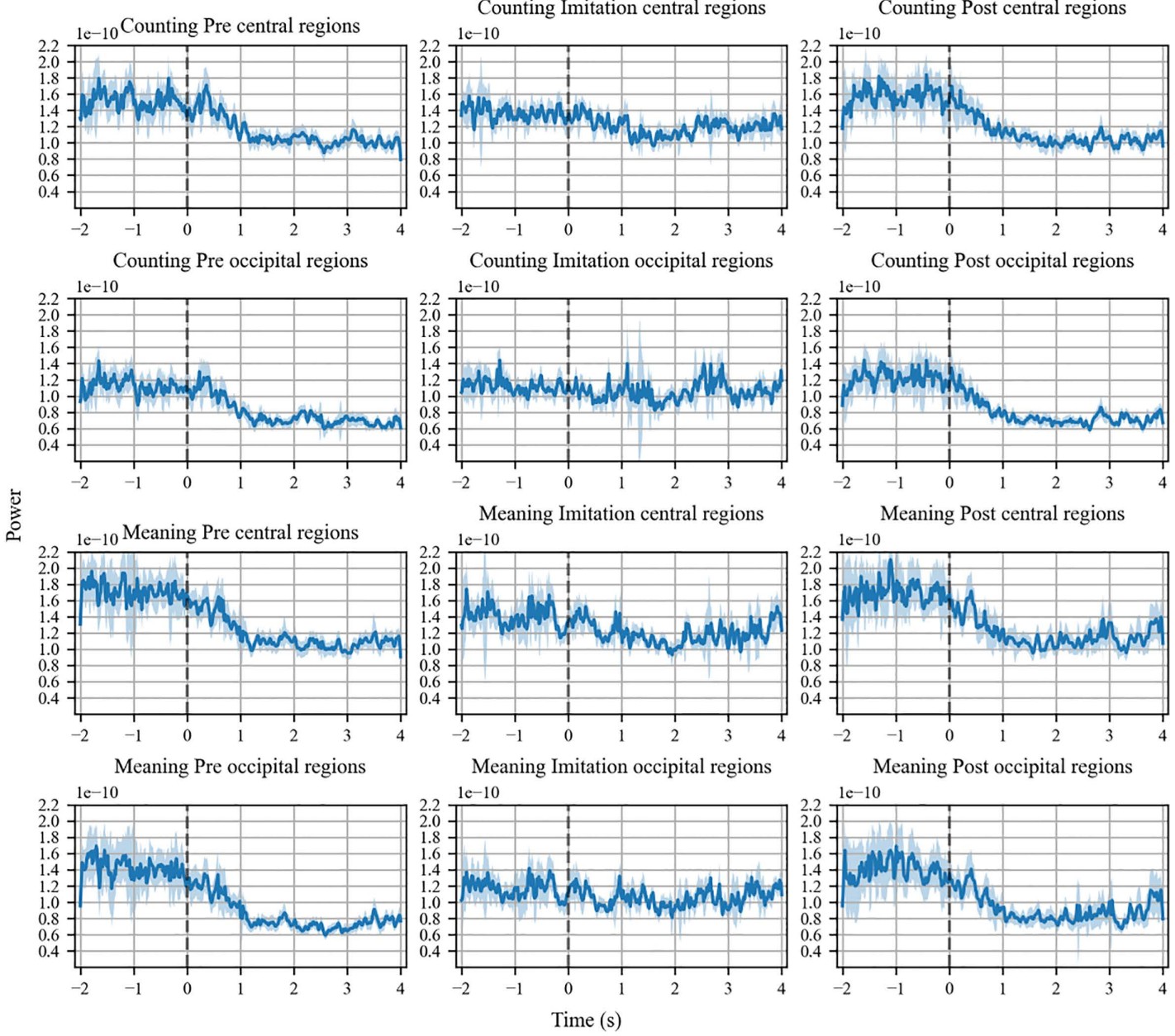

**Fig 4. Time-series mu- and alpha-power in three trial phases (pre-imitation, imitation, and post-imitation) of the counting and meaning task.**
In the counting task, distinctive mu-suppressions were found in the pre- and post-imitation phases, while the suppression was moderate in the imitation phase (1st row). Similar mu-power transition was seen in the meaning task (3rd row). Alpha-suppressions were found in the pre- and post-imitation phases while such suppression was not observed when imitating body movements in the counting (2nd row) and meaning task (4th row). The horizontal axes represent time (-2s to 4s) and the vertical ones represent power of the mu/alpha frequency. The dashed vertical lines indicate onsets of movies. The shaded area shows standard errors of the mean.

attributed to mu wave, not alpha (2nd and 4th row). S1 Fig and S2 Fig summarize power changes over time across 8–30 Hz during three phases of the counting and meaning tasks (S2 for the central regions and S3 for the occipital regions). In both the counting task and the meaning task, change of power show similar patterns. In the pre- and post-imitation phases

of the counting task, mu power began to decrease when movies started to play (0 second), and reached its plateau at 1 s after the play, which had lasted until the movies ended (4 second). In contrast, in the imitation phase, mu power slightly decreased from 1 s to 2 s, but appeared to return to the baseline level after that. Alpha power showed a similar transition to mu power in both the counting and the meaning task, but as a whole, its power was relatively smaller compared to that of mu. In the imitation phase, alpha power did not decrease throughout the phase in both tasks.

Fig 5 illustrates the proportions of mu and alpha power while participants observed or imitated sign language, relative to baseline measurements. The t-test results revealed significant changes in power ratios: counting pre-mu: $t$ (31) = −6.31, $p < 0.001$, $d = −1.58$, 95% CI [−0.28, −0.14]; counting pre-alpha: $t$ (31) = −7.68, $p < 0.001$, $d = −1.92$, 95% CI [−0.31, −0.18]; counting post-mu: $t$ (31) = −8.54, $p < 0.001$, $d = −2.13$, 95% CI [−0.29, −0.18]; counting post-alpha: $t$ (31) = −9.92, $p < 0.001$, $d = −2.48$, 95% CI [−0.34, −0.22]; meaning pre-mu: $t$ (31) = −6.68, $p < 0.001$, $d = −1.72$, 95% CI [−0.32, −0.17]; meaning pre-alpha: $t$ (31) = −8.38, $p < 0.001$, $d = −2.01$, 95% CI [−0.39, −0.24]; meaning post-mu: $t$ (31) = −7.85, $p < 0.001$, $d = −1.96$, 95% CI [−0.29, −0.17]; meaning post-alpha: $t$ (31) = −8.81, $p < 0.001$, $d = −2.20$, 95% CI [−0.32, −0.20]. These results showed that suppression occurred in both mu and alpha power as participants started to watch the signing videos.

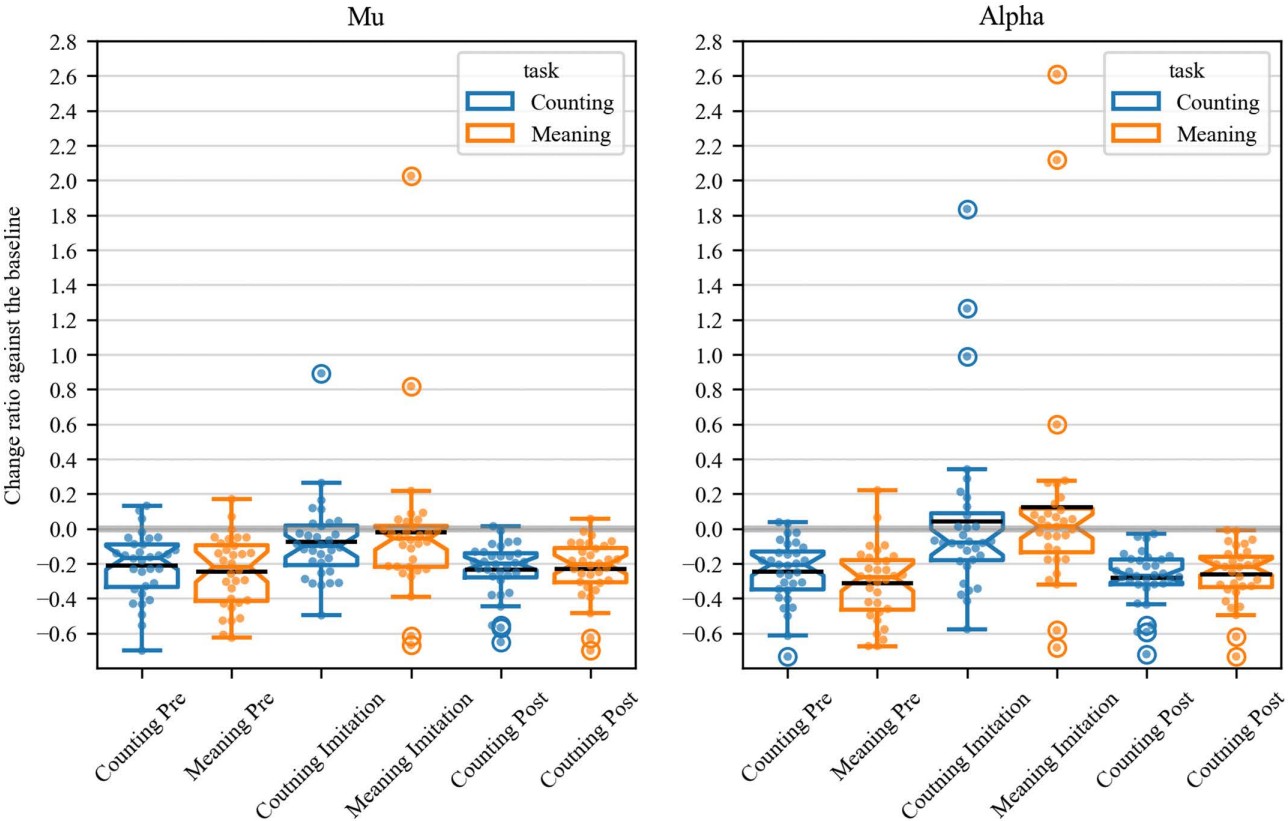

**Fig 5. Mean mu- and alpha-suppression in three trial phases of the counting (blue boxplots) and meaning task (orange box plots).** Mu-suppressions were found in the pre- and post-imitation phases in both tasks (left panel). Similarly, alpha-suppressions were observed in the pre- and post- imitation phases in both tasks (right panel). It was also found that power of mu was significantly smaller than that of alpha during the imitation phase of the meaning task. There was a trend toward a significant difference between power of mu and that of alpha during the imitation phase of the counting task. The middle lines indicate the mean values of power ratio relative to the baseline, while the notch indicates the median. The whiskers extend 1.5 times the interquartile range from the top and bottom of the box.

During the imitation phase in the counting task, the t-test showed a trend toward a significance that change in mu-power ratio was lower than zero: $t(31) = -1.79$, $p = 0.08$, $d = -0.45$, 95% CI [−0.16, 0.01]. By contrast, the t-test showed that change in mu-ratio was not significantly different from zero in the meaning task: $t(31) = -0.28$, $p = 0.78$, $d = -0.07$, 95% CI [−0.18, 0.14]. As for change in alpha-power ratio, the t-test showed that the change was significantly different from zero in neither counting task: $t(31) = 0.51$, $p = 0.62$, $d = 0.13$, 95% CI [−0.13, 0.22], nor in the meaning task: $t(31) = 1.09$, $p = 0.29$, $d = 0.27$, 95% CI [−0.11, 0.35]. Additionally, the mu and alpha power significantly differed during the imitation phase of the meaning task: $t(31) = -2.98$, $p = 0.006$, $d = -0.26$, 95% CI [−0.24, −0.05]), but not in the counting task ($t(31) = -1.91$, $p = 0.065$, $d = -0.31$, 95% CI [−0.25, 0.01]). Those results may indicate that 8–13 Hz oscillation detected in the channels above the sensorimotor cortex (C3, Cz, C4) reflect mu waves rather than alpha waves.

### Effects of the task and imitative experience on mu- and alpha-power

For all prior distributions, the Jeffreys–Zellner–Siow (JZS) [35] was used as the default option of the BayesFactor package. All combinations of conditions were examined as the alternative hypothesis, which propose that the change of power was caused by conditions' main effects or their interaction, against the null hypothesis, which proposes that all change of power could be explained as individual differences. The interpretation of the Bayes factor followed the categorical classification introduced by Schönbrodt & Wagenmakers [36], which was originally suggested by Jeffreys [37] and modified by Lee & Wagenmakers [38].

BANOVA results indicated that the effect of task anecdotally favored the null hypothesis ($BF_{10} = 0.53$, $BF_{01} = 1.87$), while the effect of practice moderately favored the null hypothesis ($BF_{10} = 0.20$, $BF_{01} = 5.06$). Finally, the model containing the effects of practice, condition, and their interaction very strongly favored the null hypothesis ($BF_{10} = 0.028$, $BF_{01} = 35.09$).

### Results of beta-power

Similar to the mu-power, beta-power of the central regions decreased when the participants observed the sign speech in the pre- and post-imitative phases across the tasks, compared with the baseline (Fig 6 1st and 3rd row). Likewise, beta-power of the occipital regions decreased during the pre- and post-imitative phases in both tasks relative to the baseline (Fig 6 2nd and 4th row). During the imitation-phase, the reduction of beta-power was found in the central regions, which was not the case with the mu-power in the central regions. By contrast, beta-power was not reduced in the occipital regions, which was similar to the alpha power in the same regions.

Fig 7 illustrates the proportions of beta power in the central (left panel) and occipital regions (right panel) while participants observed or imitated sign language, relative to baseline measurements. T-tests, which examined whether change ratio of the beta-power against the baseline was statistically different from zero in the central regions, showed significant results as follows: counting pre-beta: $t(31) = -9.43$, $p < 0.001$, $d = -2.36$, 95% CI [−0.25, −0.16]; counting post- beta: $t(31) = -11.85$, $p < 0.001$, $d = -2.96$, 95% CI [−0.27, −0.19]; meaning pre- beta: $t(31) = -8.46$, $p < 0.001$, $d = -2.12$, 95% CI [−0.27, −0.16]; meaning post- beta: $t(31) = -9.46$, $p < 0.001$, $d = -2.37$, 95% CI [−0.26, −0.17]. T-tests on the change ratio of the beta -power in the occipital regions yielded similar results: counting pre-beta: $t(31) = -12.86$, $p < 0.001$, $d = -3.21$, 95% CI [−0.29, −0.21]; counting pos-beta: $t(31) = -12.08$, $p < 0.001$, $d = -3.02$, 95% CI [−0.31, −0.22]; meaning pre-beta: $t(31) = -10.06$, $p < 0.001$, $d = -2.52$, 95% CI [−0.34, −0.22]; meaning post- beta: $t(31) = -11.20$, $p < 0.001$, $d = -2.37$, 95% CI [−0.30, −0.21].

The t-tests showed significant results on the beta-power from the central regions during the imitation phase in the counting task: $t(31) = -7.18$, $p < 0.001$, $d = -1.80$, 95% CI [−0.19, −0.11] and the meaning task: $t(31) = -3.95$, $p < 0.001$, $d = -0.99$, 95% CI [−0.201, −0.064]. By contrast, the beta-power from the occipital regions was not statistically different from zero during the imitation phase in the counting task: $t(31) = -1.75$, $p = 0.09$, $d = -0.44$, 95% CI [−0.156, 0.012] and the meaning task: $t(31) = -0.78$, $p = 0.44$, $d = -0.19$, 95% CI [−0.16, 0.07].

When comparing change ratio of beta-power between the central and occipital regions during the imitation phase in the counting task, a paired t-test showed a trend toward significant difference: $t(31) = -1.91$, $p = 0.07$, $d = -0.31$, 95% CI

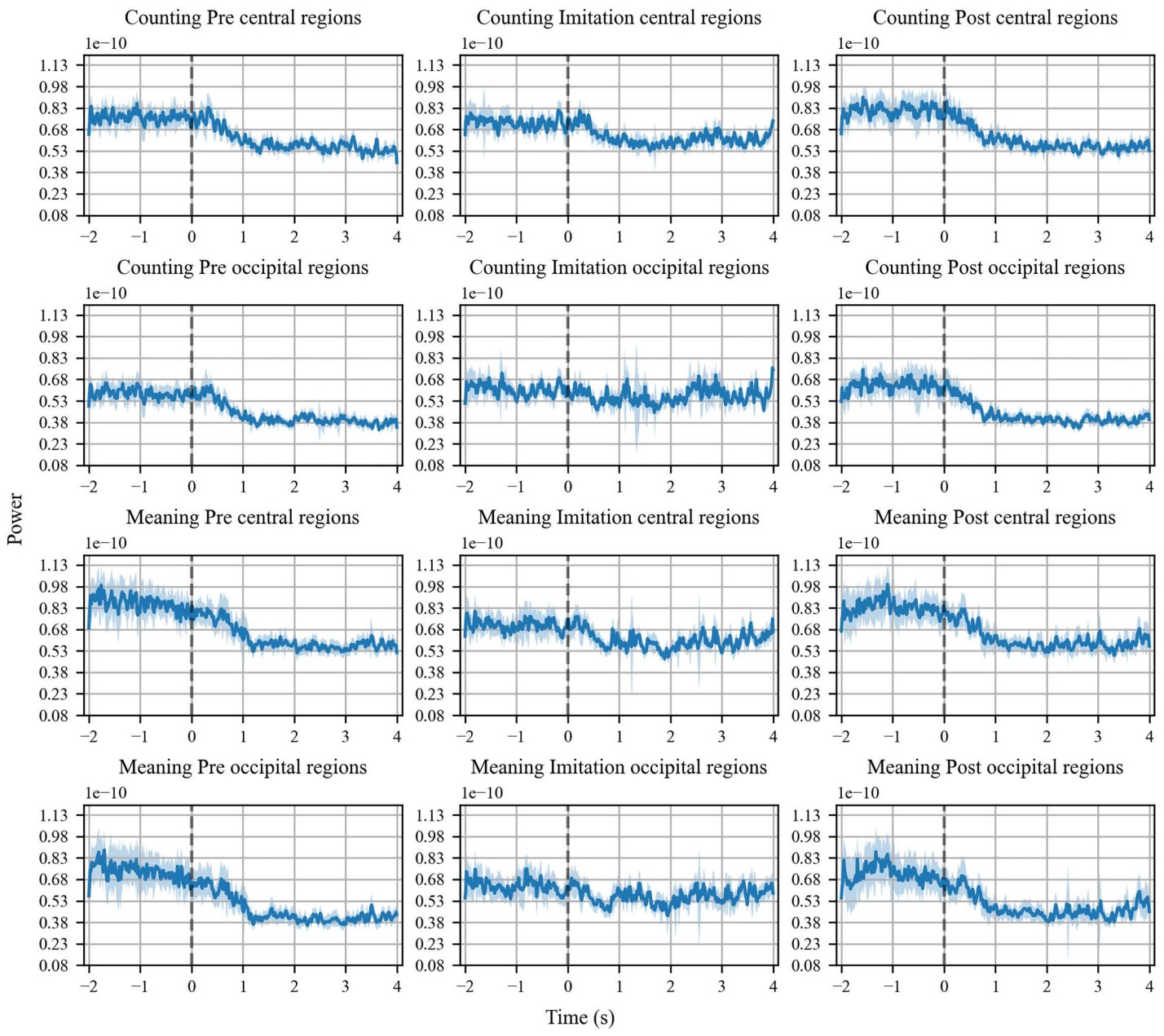

**Fig 6. Time-series beta-power in three trial phases of the counting and meaning task.** Beta-suppression in the central regions was evident in the pre- and post-imitation phases of the counting (1st row) and meaning task (3rd row), while it was moderate in the imitation phases. Beta-suppression in the occipital regions was also evident in the pre- and post-imitation phases of the counting (2nd row) and meaning task (4th row), but not observable in the imitation phases.

[−0.25, 0.01]. Moreover, the t-test showed a significantly reduced beta-power from the central regions than the occipital regions during the imitation phase of the meaning task: $t$ (31) = −2.98, $p = 0.006$, $d = −0.26$, 95% CI [−0.24, −0.05].

According to BANOVA, the effect of task moderately favored the null hypothesis ($BF_{10} = 0.28$, $BF_{01} = 3.58$). Likewise, the effect of imitation moderately favored the null hypothesis ($BF_{10} = 0.19$, $BF_{01} = 5.40$). The model containing the effects of imitation, task, and their interaction very strongly favored the null hypothesis ($BF_{10} = 0.03$, $BF_{01} = 34.44$).

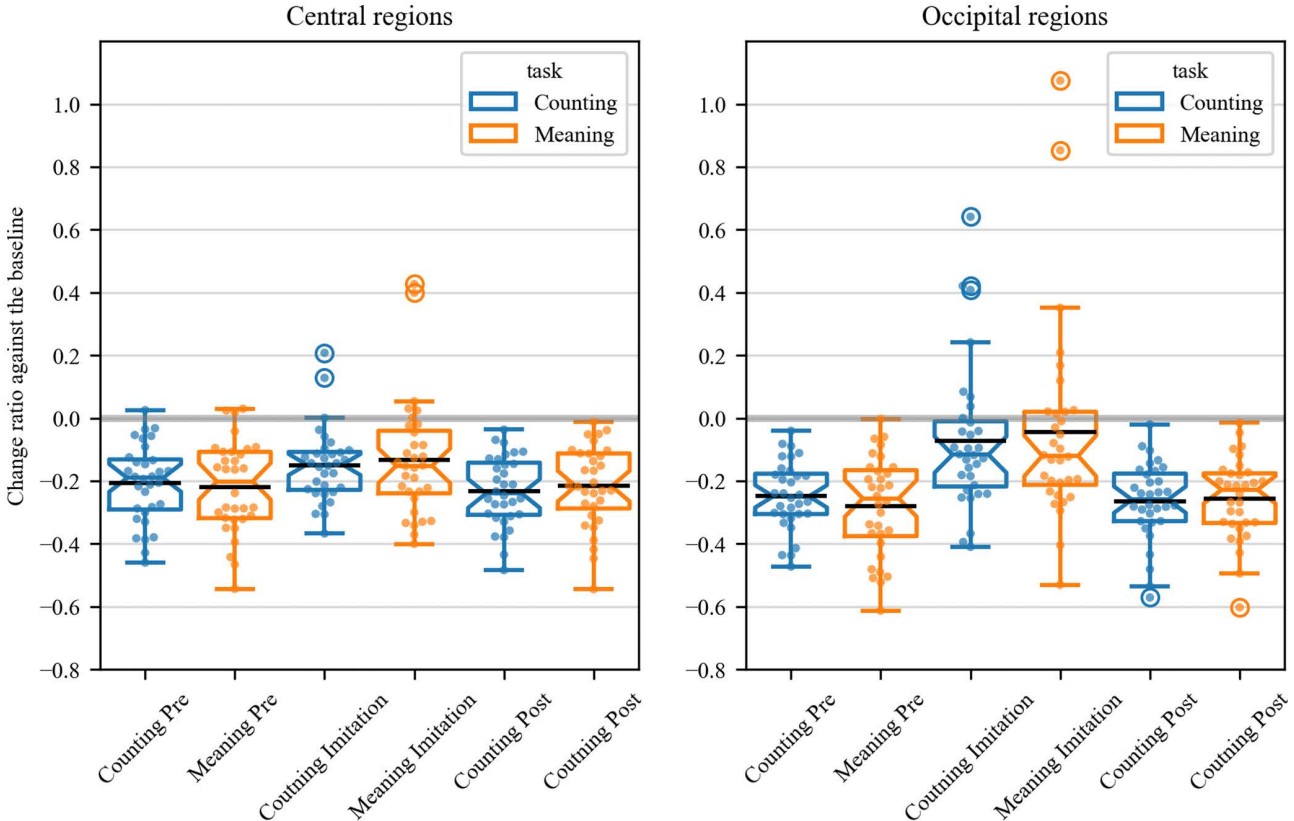

**Fig 7. Beta-suppression in three trial phases of the counting (blue boxplots) and meaning task (orange box plots).** Beta-suppression was observed in three phases in the central regions in both tasks (left panel). By contrast, the suppression was observed in the pre- and post-imitation phases but not during imitation in the occipital regions (right panel). It was also found that beta-suppression during the imitation phase of the meaning task was significantly greater in the central region than in the occipital region. The middle lines indicate the mean values of power ratio relative to the baseline, while the notch indicates the median. The whiskers extend 1.5 times the interquartile range from the top and bottom of the box.

## Results of machine learning

The mean accuracy of the target machine was 0.718, where the machine was trained to predict original labels by two features (i.e., contribution ratio of the first and second components). As for the machine trained with randomized labels, the average accuracy was 0.499. In the permutation, there were not any comparisons where the randomized machine showed higher accuracy than the target machine ($p = .002$).

The mean precision of the target machine was 0.698, whereas that of the randomized machines was 0.500. The obtained p-value by the permutation test was 0.012, indicating that the randomized machine showed higher precision than the target machine 6 times out of 500 comparisons. The mean recall of the target machine was 0.767, whereas that of the randomized machines was 0.497. The obtained p-value by the permutation test was 0.020, indicating that the randomized machine showed higher recall than the target machine 10 times out of 500 comparisons. The mean F1-score of the target machine was 0.724, whereas that of the randomized machines was 0.473. The obtained p-value by the permutation test was 0.002, indicating that there were not any comparisons where the randomized machine showed higher F1-score than the target machine.

## Discussion

The present study investigated the effects of brief imitative experience of an unfamiliar action and intention reading on the MNS. The EEG mu band was measured while participants observed or performed Japanese sign language. In the counting task, participants merely focused on the physical movements of the sign language. Conversely, the meaning task required participants to actively extract the intention of the performer based on their body movements. Given that the MNS demonstrates greater activation when understanding the intention of others [5], we hypothesized that mu suppression would be more pronounced in the meaning task than that in the counting task. We also predicted that a brief imitative experience of sign languages would enhance mu suppression, drawing on previous studies that showed mu suppression produced by active imitation of unfamiliar finger movements, tool use, and hand writing of indigenous alphabet [7]. Furthermore, we anticipated an additive effect of the two factors. That is, we postulated that the strongest mu suppression would be induced when participants tried to understand the intention of the body movements (i.e., the meaning task) which they had just imitated (i.e., the post-imitation phase).

Task performance was far above the chance level for each task, confirming that the participants were focused on the tasks. Importantly, the accuracy of the meaning task was nearly 70%, indicating that participants were moderately successful in extracting intentions from the signing actions. The result also suggests that participants seemed to pay close attention to individual sign movements and integrated these to understand the meaning of whole movements in the meaning task. Accordingly, task manipulation appeared to successfully promote intention understanding.

With respect to mu and alpha band neural oscillations, suppression in each frequency band was found during action observation. Specifically, relative to the baseline, the power of 8–13 Hz decreased in the central sites and in the occipital sites, while participants observed signing movements during both tasks (i.e., counting and meaning). Furthermore, mu and alpha band suppression was similar in the pre- and post-imitation phases. This similarity complicates the argument that the reduced power in the central sites corresponds to mu-suppression as it could also reflect alpha-related attentional processes. However, during the imitation phase, we observed a significant difference in the power between the central and occipital sites. Because the MNS was most responsive during action production [7,17,39], we might be able to capture distinctive mu suppression when participants moved their body parts rather than observing simple actions. Furthermore, the machine learning approach showed that the support vector machine was able to classify central and occipital regions above chance-level, using two feature variables (i.e., contribution ratio of the first and second components). Thus, power suppression of 8–13 Hz during sign language imitation likely reflects mu suppression rather than alpha suppression. Accordingly, it may be acceptable to consider power suppressions of the central sites in the pre- and post-imitation phases as mu suppression. However, while we found a decrease in power of 8–13 Hz oscillations in the central sites relative to the occipital sites during action imitation, there was no significant difference in power from baseline. This may be explained by artifacts induced by the real physical movement of participants, which could prevent us from measuring clean mu band oscillations. Contamination of EEG signals by physical movement were repeatedly reported [40], which could be attributed to the non-significant difference from the baseline.

To investigate an additive effect of intention reading and imitation on mu suppression, the present study employed the Bayesian analysis of variance. The result supported the null hypotheses for main effects of the task, imitation, and their interaction. Namely, mu suppressions were similar between two tasks across pre- and post-imitation phases. Therefore, mu suppression was unchanged after experiencing a brief imitation of Japanese sign language. In addition, mu suppression was similar between passive observation of body movement (i.e., counting task) and effortful observation of the movements to extract the actor's intention (i.e., meaning task).

We observed mu suppression while participants watched sign language videos, consistent with previous studies that indicate that the MNS is activated by observing unfamiliar actions [7,41]. However, we did not detect further mu wave suppression as a result of a brief imitation, contrary to the findings of Marshall et al. [7]. This discrepancy might be attributed

to the familiarity of actions. In the study by Marshall et al., writing the Cham alphabet was used as an unfamiliar movement, and a brief imitation of the movement strengthened MNS activation relative to the pre-imitation phase. However, the writing movement itself could have been familiar to participants. Conversely, movements involved in Japanese sign language may have been unfamiliar to the study participants, potentially preventing an experience effect on the MNS. Therefore, to achieve a brief imitation effect on the MNS, it may be necessary to include learned fundamental body movements even though combined actions are unfamiliar.

The account of the unfamiliar physical movement can also be applied to a null effect of the task manipulation. That is, the magnitude of mu suppression in the counting task was comparable to that of the meaning task. Because participants seemed quite unfamiliar with the physical movements associated with Japanese sign language, their MNS might be fully engaged in mirroring actions of the performer, which would make it hard for the system to read the intentions of the performer in the meaning task. If the MNS prioritizes mirroring actions over intention reading, a substantial amount of unfamiliar physical movements may consume neural resources of the MNS, thereby preventing the system from further extracting intention from the movements. It is also possible that participants could count on their knowledge to retrieve intention from unfamiliar physical movements in the tasks. Because those physical movements were not built in their action repertoire, they could infer intention of the movements with their semantic knowledge. If so, neural systems other than the MNS could be recruited in the meaning task. Knowledge-based action understanding is reported to activate traditional language and auditory areas rather than the MNS [13]. Taken together, the MNS may not be activated for intention reading if the observed actions are far away from observers' action repertoire.

When referring to studies with American Sign Language (ASL), deaf signers showed less activation of the MNS while observing ASL [11]. Conversely, hearing non-signers showed remarkable activation during ASL observation. The weak or null activity of the MNS in deaf signers seems to be attributable to knowledge-based processing of ASL rather than mirroring actions associated with ASL [42]. Recruiting deaf-signers and hearing non-signers, Kubicek and Quandt [12] measured mu suppression while observing ASL. Mu suppression was found to be greater in the hearing non-signers than in the deaf-signers. The authors also found that the complexity of ASL differentially affected MNS activity in the two groups. Specifically, hearing non-signers showed greater mu suppression in the simple actions (i.e., one-hand ASL) than the complex ones (i.e., two-hands ASL). The opposite pattern was observed in the deaf-signers. Based on the aforementioned findings, the authors suggested that MNS activity depends, to an extent, on how much observers are able to imitate the action. Because our participants were instructed to imitate the sign languages during the imitation phase, they may have prepared for imitation during the pre-imitation phase, which could have masked differences in mu suppression between the counting and meaning tasks. However, the post-imitation phase did not involve subsequent imitation, suggesting that the imitation preparation account may not apply to the null effect of the task on mu suppression during this phase.

Furthermore, it is possible that content associated with an action could differentially affect the MNS activity. In a study by Marshall et al. [7], participants observed an unfamiliar hand-writing letter in each trial, for a duration of 4–7 s ($M = 5.5$ s). As the present study drew on the research by Marshall et al. to investigate the effect of brief imitative experience on the MNS, we employed stimuli with similar duration (4.30–6.33 s). However, our participants were required to observe several sign items to form a single sentence. This difference may be the main cause for the inconsistent results between the two studies regarding the effect of brief imitative experience on mu suppression. In the study by Kubicek and Quandt [12] using ASL, each sign represented a single word, and hearing non-signers showed significant mu suppression. Although the mean duration of sign movies was less than 2 s, which was far shorter than movie stimuli described in the study by Marshall et al. [7], actions associated with simple content may be more susceptible to a brief imitative experience, resulting in enhanced mu suppression. As participants in the present study were required to observe and imitate several actions in each trial, the MNS neural resources may have been depleted, which might hinder the ability of a brief action experience to intensify mu suppression.

When using sign language to study the MNS, we may need to consider the possibility that cultural factors play an important role, particularly in terms of the frequency of gestural communication. It has been noted that usage of large and expressive gestures is strongly encouraged during oral communication in some cultures, such as Latin and Middle Eastern cultures; conversely, in other cultures, such as East Asian cultures, including Japan, expressive gestural communication is discouraged [43]. Considering that the body movements of sign language are mostly similar to body gestures used to express one's emotions or to support speech, an imitation effect of sign language on the MNS may be more evident in cultures in which body gestures are usually used. The fact that participants in this study were less likely to use body gestures in daily communications, may have removed the imitation effect on mu suppression. If this assumption is correct, it is predicted that a brief imitative experience of sign language would induce greater MNS activation in cultures where gestural communication is widely accepted. Collectively, the results of this study suggest that higher familiarity or sufficient acquisition of body movements may be a key factor yielding an imitation effect on the MNS during the observation of the body movements.

### Alpha wave and mu wave

During the observation phases, both alpha and mu wave showed decreases in power, indicating that the present tasks evoked not only mu suppression but also alpha suppression.

We attributed alpha suppression to changes in participants' focus during the tasks. Unlike mu power, which decreases when individuals observe or perform actions, alpha power has been repeatedly shown to decrease when individuals focus their attention on a target stimulus or task [18,20]. Because participants were required to count the number of movements or read the intention of a performer during the sign language tasks, they needed to pay greater attention to an action the performer made in a movie. As the onset of the alpha suppression synchronized with the onset of the movie, suppression seems to be linked to attentional engagement.

Theoretically, in the imitation phase, only mu waves show suppression but not alpha waves; therefore, mu and alpha power should significantly differ. This pattern could be observed in the meaning task. Previous studies showed that mu waves were suppressed most while participants were moving their bodies [7,39]; however, in the present study, suppression during the imitation appeared to be smaller relative to the pre- and post-imitation phases. This may be because imitating sign languages required relatively larger movements, compared with those in previous studies (e.g., grabbing the manipulandum, opening and closing palms). Greater physical movements could induce strong artifacts, which could mask mu suppression produced by the MNS.

### Beta wave

Similar to mu- and alpha-oscillation, beta-oscillation in the central regions was suppressed during pre- and post-imitative phases in both tasks. The finding is consistent with previous literature showing event-related desynchronization of the alpha- and beta-band while participants observed aimless middle-finger extension produced by other person [24]. However, beta suppression was also found in the occipital regions, which does not allow us to conclude that beta suppression in the central regions solely reflects the MNS related activity.

Analogous to mu power, beta power was reported to be more suppressed when executing action than observing [24]. Given that, difference in beta suppression associated with the MNS activity may be more detectable by comparing suppression of beta power during action execution. Because beta power was suppressed in the central but not in the occipital regions during the imitation phases, we seem to succeed in capturing neurophysiological activity linked to the MNS.

While beta suppression in the central regions may correspond to excitation of the MNS, the suppression was not modulated by the task type, imitative experience, and their interaction. Those results are consistent with the finding of the mu-suppression, indicating neither effortful intention reading nor brief imitation of unfamiliar sign language modulated neural activity of the MNS in the present study.

## Limitations and the future directions

The task order was fixed so the counting task always preceded the meaning task. As a result, participants could become less attentive in performing the meaning task due to fatigue. The procedure may have masked the effect of intention reading on mu suppression, which might have been observed otherwise. A fixed order of tasks was established to prevent participants from reading the intentions of sign language in the counting task, which they might have done if they had experienced the meaning task first. Further research is required to assess the optimal order of the task.

Additionally, body movements during imitation varied across participants with some performing large body movements during imitation and other performing small movements. Because artifacts associated with body movement seem to distort EEG data, future studies need to consider how to guide participants to make minimum body movements during imitation.

## Conclusion

In conclusion, the results showed that mu suppression was similarly evoked when participants passively viewed signing and when they effortfully read intentions embedded in the signs. In addition, mu suppression during action observations was unchanged before and after imitating the actions. The null effect of intention reading may be attributed to knowledge-based inference to retrieve intention from sign language actions, because those actions were hardly developed as their motion repertoire. The absence of the imitative experience may be due to cultural factors. That is, less use of body gestures during public communications in Asian cultures. Future studies should explore the effects of familiarity of body movements and cultural factors, which could modulate the MNS response.

## Supporting information

**S1 Movie. A sample movie of a Japanese sign speech.** In the movie, the first author performed a sign speech meaning "Long time no see, everyone. How are you?", following a book titled "Simple sign language phrases for beginners".
(MP4)

**S1 Fig. Power changes in the central regions over time across 8–30 Hz during three phases of the counting and meaning tasks.** The left columns show time-frequency plots in the counting task and the right ones show the plots in the meaning task (Top: pre-imitation, Middle: imitation, Bottom: post-imitation).
(TIF)

**S2 Fig. Power changes in the occipital regions over time across 8–30 Hz during three phases of the counting and meaning tasks.** The layout of the figures are identical to S1 Fig.
(TIF)

## Author contributions

**Conceptualization:** Tomoki Osaki, Takehiro Minamoto.

**Data curation:** Tomoki Osaki.

**Formal analysis:** Tomoki Osaki.

**Funding acquisition:** Takehiro Minamoto.

**Investigation:** Tomoki Osaki, Takehiro Minamoto.

**Methodology:** Tomoki Osaki, Takehiro Minamoto.

**Project administration:** Tomoki Osaki, Takehiro Minamoto.

**Resources:** Tomoki Osaki, Takehiro Minamoto.

**Software:** Tomoki Osaki.

**Supervision:** Takehiro Minamoto.

**Validation:** Tomoki Osaki, Takehiro Minamoto.

**Visualization:** Tomoki Osaki.

**Writing – original draft:** Tomoki Osaki, Takehiro Minamoto.

**Writing – review & editing:** Tomoki Osaki, Takehiro Minamoto.

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
