## [Decision Letter · Decision Letter 0]

5 May 2025

Dear Dr. Minamoto,

Thank you for submitting your manuscript to PLOS ONE. After careful consideration, we feel that it has merit but does not fully meet PLOS ONE’s publication criteria as it currently stands. Therefore, we invite you to submit a revised version of the manuscript that addresses the points raised during the review process.

We look forward to receiving your revised manuscript.

Kind regards,

Luigi Cattaneo, MD, PhD

Academic Editor

PLOS ONE

**Journal Requirements:**

1. When submitting your revision, we need you to address these additional requirements. Please ensure that your manuscript meets PLOS ONE's style requirements, including those for file naming. The PLOS ONE style templates can be found at https://journals.plos.org/plosone/s/file?id=wjVg/PLOSOne_formatting_sample_main_body.pdf and https://journals.plos.org/plosone/s/file?id=ba62/PLOSOne_formatting_sample_title_authors_affiliations.pdf  2. Thank you for stating the following financial disclosure: Japan Society for Promotions of Science#21K03130 and 24K00507  Please state what role the funders took in the study.  If the funders had no role, please state: "The funders had no role in study design, data collection and analysis, decision to publish, or preparation of the manuscript." If this statement is not correct you must amend it as needed. Please include this amended Role of Funder statement in your cover letter; we will change the online submission form on your behalf. 3. Thank you for stating the following in the Acknowledgments Section of your manuscript: This study was supported by research grant from Japan Society for Promotions of Science to TM (#21K03130 and 24K00507). We note that you have provided funding information that is not currently declared in your Funding Statement. However, funding information should not appear in the Acknowledgments section or other areas of your manuscript. We will only publish funding information present in the Funding Statement section of the online submission form. Please remove any funding-related text from the manuscript and let us know how you would like to update your Funding Statement. Currently, your Funding Statement reads as follows: Japan Society for Promotions of Science#21K03130 and 24K00507  Please include your amended statements within your cover letter; we will change the online submission form on your behalf. 4. We note that you have indicated that there are restrictions to data sharing for this study. For studies involving human research participant data or other sensitive data, we encourage authors to share de-identified or anonymized data. However, when data cannot be publicly shared for ethical reasons, we allow authors to make their data sets available upon request. For information on unacceptable data access restrictions, please see http://journals.plos.org/plosone/s/data-availability#loc-unacceptable-data-access-restrictions.  Before we proceed with your manuscript, please address the following prompts: a) If there are ethical or legal restrictions on sharing a de-identified data set, please explain them in detail (e.g., data contain potentially identifying or sensitive patient information, data are owned by a third-party organization, etc.) and who has imposed them (e.g., a Research Ethics Committee or Institutional Review Board, etc.). Please also provide contact information for a data access committee, ethics committee, or other institutional body to which data requests may be sent. b) If there are no restrictions, please upload the minimal anonymized data set necessary to replicate your study findings to a stable, public repository and provide us with the relevant URLs, DOIs, or accession numbers. Please see http://www.bmj.com/content/340/bmj.c181.long for guidelines on how to de-identify and prepare clinical data for publication. For a list of recommended repositories, please see https://journals.plos.org/plosone/s/recommended-repositories. You also have the option of uploading the data as Supporting Information files, but we would recommend depositing data directly to a data repository if possible. Please update your Data Availability statement in the submission form accordingly. 5. We note that Figure 1 includes an image of a participant in the study. As per the PLOS ONE policy (http://journals.plos.org/plosone/s/submission-guidelines#loc-human-subjects-research) on papers that include identifying, or potentially identifying, information, the individual(s) or parent(s)/guardian(s) must be informed of the terms of the PLOS open-access (CC-BY) license and provide specific permission for publication of these details under the terms of this license. Please download the Consent Form for Publication in a PLOS Journal (http://journals.plos.org/plosone/s/file?id=8ce6/plos-consent-form-english.pdf). The signed consent form should not be submitted with the manuscript, but should be securely filed in the individual's case notes. Please amend the methods section and ethics statement of the manuscript to explicitly state that the patient/participant has provided consent for publication: “The individual in this manuscript has given written informed consent (as outlined in PLOS consent form) to publish these case details”.  If you are unable to obtain consent from the subject of the photograph, you will need to remove the figure and any other textual identifying information or case descriptions for this individual.

Reviewers' comments:

Reviewer's Responses to Questions

**Comments to the Author**

1. Is the manuscript technically sound, and do the data support the conclusions?

Reviewer #1: Partly

Reviewer #2: Partly

2. Has the statistical analysis been performed appropriately and rigorously?

Reviewer #1: I Don't Know

Reviewer #2: Yes

3. Have the authors made all data underlying the findings in their manuscript fully available?

Reviewer #1: No

Reviewer #2: Yes

4. Is the manuscript presented in an intelligible fashion and written in standard English?

Reviewer #1: Yes

Reviewer #2: No

**Reviewer #1:**  The study by Minamoto and Colleagues investigates the activation of the mirror neuron system (MNS) during the observation and imitation of Japanese Sign Language, an unfamiliar action for the participants. By analyzing mu rhythm suppression in EEG signals, the study evaluates the effects of intentionality on gesture comprehension and the impact of practice on MNS activity. Understanding how the human brain processes unfamiliar gestures can provide valuable insights into the mechanisms of learning and social cognition. However, several aspects in the description of the methods and results reduce my enthusiasm for the findings presented in this manuscript.

-Sample size: The study includes 37 participants, but the final EEG data is based on only 32 subjects due to the exclusion of some data for artifacts. While this number is not excessively low, it may be useful to discuss the statistical power of the study.

-Mu vs. alpha comparison: The study attempts to distinguish mu rhythm suppression from alpha wave suppression, but the chosen method (analyzing central channels for mu and occipital channels for alpha) may not be sufficient to rule out overlap between the two components. I do not believe that an in-depth discussion alone is enough on this point; the authors should consider using more sophisticated methodologies to separate these rhythms, such as Independent Component Analysis (ICA).

-Randomized trial order: Was the order of trials (counting task first, meaning task afterward) randomized?

-EEG baseline selection: The baseline was defined as the first static frame of the video. However, using a static image may not be an ideal control. How can it be ruled out that participants do not anticipate the movement and already exhibit different neural activation compared to a true resting state? I would suggest adding control analyses to justify this choice.

-Morlet Wavelet parameters: The key parameters of the Morlet wavelet transform, such as the number of cycles used, are not specified. This is a critical aspect, as a low number of cycles provides good temporal resolution but poor frequency resolution, whereas a high number has the opposite effect. Furthermore, I would have explored beta rhythms as well, given their functional association with the motor system.

-Electrodes: It is necessary to precisely indicate the sensors used for the analyses, referring to the standard nomenclature (e.g., O1, O2, etc.). Additionally, providing a template with the spatial mapping of the selected electrodes would be beneficial.

-EEG data figures: The current figures do not clearly allow an appreciation of signal quality or inter-subject variability. It would be useful to plot individual values. Moreover, since temporal resolution is one of EEG's strengths, it would also be helpful to include figures showing power changes over time across a broader frequency range. Classic wavelet plots are very useful tools for visualizing the variation in the power of a signal over time and across different frequencies.

**Reviewer #2:**  In the study entitled “Effects of Intention Understanding and Action Practice on the Mirror Neuron System: An EEG Study using Japanese Sign Language”, the authors investigate how two cognitive processes—intention understanding and action practice—affect activation of the mirror neuron system (MNS), as indexed by mu suppression measured through EEG. While the study addresses a relevant question in social neuroscience, I find that the current version suffers from significant conceptual and methodological limitations.

The term “practice” is used throughout the manuscript (e.g., pre-practice, practice, post-practice) to describe participants’ brief imitation of sign language gestures. However, I find the use of this term misleading. Based on the experimental design, participants first observe an unfamiliar gesture (pre-practice), then imitate it once (practice), and finally observe the same gesture again (post-practice). Simply observing a movement multiple times does not equate to motor practice, particularly for symbolic, unfamiliar movements such as those in sign language.

The hypothesis that participants engaged in “intention understanding” during the meaning task is problematic. The authors appear to conflate intention inference with semantic recognition. In the context of sign language, interpreting a gesture generally involves identifying its linguistic meaning, not inferring the actor’s motor goal. In mirror neuron research, "intention understanding" refers to recognizing the purpose behind a motor action (e.g., grasping to drink vs. to move)—not mapping symbolic gestures to predefined meanings. Instructing participants to choose the correct meaning from a list of sentences is a semantic task, not a test of motor intention. In my view, the study is better characterized as investigating semantic processing or gesture recognition, rather than “intention understanding” as defined in the MNS literature.

In the Introduction and early parts of the Discussion, the authors heavily rely on findings from a study involving the Cham alphabet to support the claim that brief motor practice of unfamiliar actions enhances MNS activity. However, this study is neither cited nor included in the reference list. Given that it appears to be a key source underpinning the theoretical rationale for the experiment, it is essential that the authors (1) provide a complete and accurate citation for this study, and (2) clarify whether its findings genuinely support the mechanisms proposed (e.g., effects of unfamiliar symbolic content, MNS modulation via brief motor exposure).

Furthermore, the use of references 6 (Cannon et al.) and 7 (Marshall et al.) is problematic. The Cannon et al. study does not focus on unfamiliar actions, nor does it show that observing and then practicing such actions enhances MNS activity. On the contrary, it emphasizes that self-generated motor experience is a stronger driver of mu desynchronization than observational learning. The Marshall et al. study cited does not investigate practice effects or unfamiliar action learning; rather, it reports overlapping EEG responses to observed and executed actions in infants. These references do not support the claim that brief practice of unfamiliar actions enhances MNS activity, and the sentence in question should be revised or removed accordingly.

Given these significant conceptual concerns, I believe it is premature to evaluate the EEG methodology and findings. It is essential that the authors first clarify their operational definitions of “practice” and “intention understanding,” justify the use of these terms within the framework of MNS research, and revise their theoretical rationale and citations accordingly.

**Do you want your identity to be public for this peer review?** For information about this choice, including consent withdrawal, please see our Privacy Policy

Reviewer #1: No

Reviewer #2: No

---

## [Author Response · Author response to Decision Letter 1]

18 Aug 2025

Dear Reviewer1,

We are mostly grateful for your thoughtful suggestions that will substantially improve our manuscript. We seriously took your suggestions into consideration, and reflected them on the revised manuscript. We hope the revised manuscript successfully addressed your concerns.

Comment 1

Sample size: The study includes 37 participants, but the final EEG data is based on only 32 subjects due to the exclusion of some data for artifacts. While this number is not excessively low, it may be useful to discuss the statistical power of the study.

Response

We appreciate the reviewer's comment on statistical power of the data analysis. In the present study, we employed the Bayesian analysis of variance to test effects of the task and imitative experience on neural oscillation's data. It is suggested that "power" is defined from a frequentist perspective, but is not directly applicable to the Bayesian framework (Schmalz et al., 2023). This is because power is the long-run probability of finding an effect, which is inherent to frequentist concept. As the Bayesian analyses employ priors provided by the precedent studies, and obtained results will be embedded into their priors for the following studies, the concept of power cannot be directly applied.

Although the present study did not use a repeated ANOVA for our analysis, we ran the power analysis (effect size = 0.25, α = .05, β = 0.8, number of measurements = 4) to compute sample size required to draw meaningful conclusion. The analysis recommended a sample size of 24, which is smaller than the size of the present study.

Those issues are addressed in Method (pp.14-15).

Comment 2

Mu vs. alpha comparison: The study attempts to distinguish mu rhythm suppression from alpha wave suppression, but the chosen method (analyzing central channels for mu and occipital channels for alpha) may not be sufficient to rule out overlap between the two components. I do not believe that an in-depth discussion alone is enough on this point; the authors should consider using more sophisticated methodologies to separate these rhythms, such as Independent Component Analysis (ICA).

Response

Acknowledging the reviewer's suggestion, we introduced additional analysis to test whether mu- and alpha-rhythm were dissociated. For the analysis, we employed the independent component analysis (ICA) and a machine learning method (i.e., support vector machine). Details of the analyses are described in Method (pp.16-18). The result of the permutation test showed that the trained machine successfully distinguished the mu-rhythms from the alpha (pp.14), supporting our assumption that the neurophysiological signal measured from the central regions reflects neural the MNS-related activity.

Comment 3

Randomized trial order: Was the order of trials (counting task first, meaning task afterward) randomized?

Response

We should have provided trial orders in the previous manuscript. All the participants performed the counting task first to prevent a risk that prior experience of the meaning task could encourage the participants to infer the meaning of sign languages while counting number of body movements. In each task, stimulus order was randomized across the participants. Those issues are addressed on pp.10-11.

Comment 4

EEG baseline selection: The baseline was defined as the first static frame of the video. However, using a static image may not be an ideal control. How can it be ruled out that participants do not anticipate the movement and already exhibit different neural activation compared to a true resting state? I would suggest adding control analyses to justify this choice.

Response

We should have provided a rationale to employ the first static frame as the baseline in the original manuscript. Hobson and Bishop (2016) tested several baseline measurements to capture mu-suppression in studies of the MNS. The authors suggested that the initial static frame of each movie stimulus can be used as the most appropriate baseline relative to others such as a cross fixation. The static initial frame is less likely to produce a change of attentional state due to smooth transition to the movie, while others (e.g., cross fixation) elevate attentional level because of grater shift of visual input. We clearly provided the rationale on pp.14.

Comment 5

Morlet Wavelet parameters: The key parameters of the Morlet wavelet transform, such as the number of cycles used, are not specified. This is a critical aspect, as a low number of cycles provides good temporal resolution but poor frequency resolution, whereas a high number has the opposite effect. Furthermore, I would have explored beta rhythms as well, given their functional association with the motor system.

Response

We should have clearly described the number of cycles to be used for the wavelet analysis. We used the default value (i.e., 2 cycles) given by Python-MNE. Because the value is low, the obtained results appear to have better temporal resolution but may suffer from lower frequency resolution. We addressed the issue on pp.13.

Following the reviewer’s suggestion, we analyzed beta rhythms and provided the results on pp21-24 and our interpretation on the results in Discussion on pp.31.

Comment 6

Electrodes: It is necessary to precisely indicate the sensors used for the analyses, referring to the standard nomenclature (e.g., O1, O2, etc.). Additionally, providing a template with the spatial mapping of the selected electrodes would be beneficial.

Response

Following the reviewer’s suggestion, we added the standard sensor labels to our target channels on pp.11-12. We also provide Figure 2 showing a spatial layout of the sensors.

Comment 7

EEG data figures: The current figures do not clearly allow an appreciation of signal quality or inter-subject variability. It would be useful to plot individual values. Moreover, since temporal resolution is one of EEG's strengths, it would also be helpful to include figures showing power changes over time across a broader frequency range. Classic wavelet plots are very useful tools for visualizing the variation in the power of a signal over time and across different frequencies.

Response

Appreciating the reviewer's comment, we added shade to lines graphs (Figure 2 in the initial manuscript) to show standard error of the mean in each time point, which is labeled as Figure 4 in the revised manuscript. We also replaced bar graphs (Figure 3 in the initial manuscript) with box and whisker plots (Figure 5). Additionally, we provide the supplementary figures (S2 and S3 Figure) summarizing power changes over time across 8-30Hz during three phases of the counting and meaning tasks.

Dear Reviewer2,

We are mostly grateful for your thoughtful suggestions as for the conceptual aspects of the study framework, which are quite important to sophisticate our manuscript. We seriously took your suggestions into consideration, and reflected them on the revised manuscript. We hope the revised manuscript successfully addressed your concerns.

Comment 1

The term “practice” is used throughout the manuscript (e.g., pre-practice, practice, post-practice) to describe participants’ brief imitation of sign language gestures. However, I find the use of this term misleading. Based on the experimental design, participants first observe an unfamiliar gesture (pre-practice), then imitate it once (practice), and finally observe the same gesture again (post-practice). Simply observing a movement multiple times does not equate to motor practice, particularly for symbolic, unfamiliar movements such as those in sign language.

Response

Agreeing with the reviewer’s opinion, we replaced the term “practice” with “imitative experience”, which was used in the previous study (Marshall et al., 2009, Neuropsychologia), throughout the manuscript.

Comment 2

The hypothesis that participants engaged in “intention understanding” during the meaning task is problematic. The authors appear to conflate intention inference with semantic recognition. In the context of sign language, interpreting a gesture generally involves identifying its linguistic meaning, not inferring the actor’s motor goal. In mirror neuron research, "intention understanding" refers to recognizing the purpose behind a motor action (e.g., grasping to drink vs. to move)—not mapping symbolic gestures to predefined meanings. Instructing participants to choose the correct meaning from a list of sentences is a semantic task, not a test of motor intention. In my view, the study is better characterized as investigating semantic processing or gesture recognition, rather than “intention understanding” as defined in the MNS literature.

Response

As the reviewer pointed out, the meaning task in the present study required inferences of action meaning (i.e., what the actor's movement meant) rather than intention understanding (i.e., why the actor made that movement). We agree with the reviewer's comment that intention understanding is a process to infer a goal or motivation behind certain body movements. Regarding with the direct perception account of intention understanding by the MNS, perception of other's body movements has the observer immediately attribute an intentional meaning to the movements (Rizzolatti and Sinigaglia,2007). This process may hold true for gestural communication. For instance, Montgomery et al. (2007) measured activity of the mirror neuron areas (i.e., inferior parietal lobule and frontal operculum) while participants were viewing, imitating, and producing object-directed hand movements or communicative hand gestures. When compared with the rest period (i.e., viewing a blank screen), the IPL and frontal operculum showed greater activation in both movements. Moreover, the activations of the IPL and the frontal operculum were quite similar between the movements. That is, the activities in the imitation and production phases were greater than the view phase in both movements, and their time-course transition of percent signal change showed high resemblance. The result seems to indicate that gestural movement for social communication activates the MNS to grasp intentional meaning of the other actors, without target-object for body movements. In monkey, mouth movements for communication, which were not object-directed, fired mirror neurons in area F5 (Ferrari et al., 2003). The result also seems to suggest that communicative body movements, including gestures, activate the MNS to read intentions that an actor tries to express in addition to superficial meaning of the gestures. Interestingly enough, when comparing the MNS-related activity between deaf signers and hearing non-signers while they were observing pantomimes or sign languages, studies have found less activity in the former groups than the latter (Emmorey et al. 2010; Kuibicek and Quandt, 2019). Instead, deaf-signers showed greater activity in the left hemisphere language areas including the inferior frontal gyrus and superior temporal sulcus (Newman et al., 2015). Those results indicate that deaf signers are likely to depend on semantic processing when perceiving others' manual movements. By contrast, hearing non-signers appear to count on the MNS when perceiving the movements. Collectively, we assume that hearing non-signers would show the MNS-related neural activity when they view sign-languages, which is not only for others' motion perception but also for reading their intentions behind body movements.

Adding the paragraph above to Introduction (pp.5-7), we clarify how intentional understanding is involved in the meaning task of the present study. We would like to welcome further comments by the reviewer to overcome weaknesses of the theoretical framework in the revised manuscript.

Comment 3

In the Introduction and early parts of the Discussion, the authors heavily rely on findings from a study involving the Cham alphabet to support the claim that brief motor practice of unfamiliar actions enhances MNS activity. However, this study is neither cited nor included in the reference list. Given that it appears to be a key source underpinning the theoretical rationale for the experiment, it is essential that the authors (1) provide a complete and accurate citation for this study, and (2) clarify whether its findings genuinely support the mechanisms proposed (e.g., effects of unfamiliar symbolic content, MNS modulation via brief motor exposure).

Response

Appreciating the reviewer's comment, we checked the original manuscript and found that we cited the wrong article. We replaced the Marshall et al.'s paper (2011) in the Developmental Science with the Marshall et al.'s paper (2009) in the Neuropsychologia in the revised manuscript. In addition to the Marshall et al's study (2009), we cited related literature (Brunsdon et al., 2019) and rewrote the paragraph as follows:

Studies have shown that active experience of unfamiliar action enhances MNS activity when the person subsequently observed that action. For instance, Brunsdon et al. (2019) found that imitations of unfamiliar finger movements or tool use enhanced mu-suppression while observing the same actions subsequently (Brunsdon et al. 2019). In the study, the participants wore the EEG cap and observed actions presented on the monitor prior to an experimental manipulation. Then, half of the participants imitated the target actions while watching the video clips. The other half observed the video clips. The authors also manipulated repetitions of the imitation or observation. After the procedure, all the participants observed the actions again. The result showed that mu-desynchronization was greater in the imitation condition than the observation condition. They also found a null effect of the repetitions, that is, mu-desynchronization was greater in the imitation condition regardless of number of repetitions. Those results indicate that brief imitation of an unfamiliar action makes the MNS more responsive when the same action was observed. It was also shown that brief imitation of an unfamiliar writing action produced greater suppression of mu-rhythm at the central channels, suggesting that activity of the MNS involves in the early stages of imitative learning (Marshall et al., 2009). In the study, participants observed unfamiliar writing movements of the Cham alphabet, which is used by an ethnic group of Southeast Asia. Under the experimental condition, participants were asked to: first, observe the writing of the Cham alphabet; second, imitate this writing; third, observe the writing again; lastly, imitate the writing again. Under the control condition, 50% of all trials were set as the “other motor experience” condition, which required participants to write two English letters after the first observation of writing the Cham alphabet, and imitate writing the Cham alphabet after the second observation. EEG recordings were acquired during the first and the second observation, and mu suppression was calculated. The results showed greater mu wave suppression under the experimental condition relative to the control condition. The two studies described above may indicate that brief motor experience of unfamiliar actions allow the MNS to embed the actions into its motor repertoire, which can be mirrored subsequently to facilitate the action observation.

The paragraph above was placed on pp.3-4 in the revised manuscript.

Comment 4

Furthermore, the use of references 6 (Cannon et al.) and 7 (Marshall et al.) is problematic. The Cannon et al. study does not focus on unfamiliar actions, nor does it show that observing and then practicing such actions enhances MNS activity. On the contrary, it emphasizes that self-generated motor experience is a stronger driver of mu desynchronization than observational learning. The Marshall et al. study cited does not investigate practice effects or unfamiliar action learning; rather, it reports overlapping EEG responses to observed and executed actions in infants. These references do not support the claim that brief practice of unfamiliar actions enhances MNS activity, and the sentence in question should be revised or removed accordingly.

Response

Following the plausible suggestion, we removed two references, and rewrote the par

---

## [Decision Letter · Decision Letter 1]

13 Oct 2025

Dear Dr. Minamoto,

Thank you for submitting your manuscript to PLOS ONE. After careful consideration, we feel that it has merit but does not fully meet PLOS ONE’s publication criteria as it currently stands. Therefore, we invite you to submit a revised version of the manuscript that addresses the points raised during the review process.

as you will see the reviewer was fully satisfied with the revision work. Please address the minor suggestions in a revised version. Once this is done, the new version will not be sent off for further review.

We look forward to receiving your revised manuscript.

Kind regards,

Luigi Cattaneo, MD, PhD

Academic Editor

PLOS ONE

Journal Requirements:

Reviewers' comments:

Reviewer's Responses to Questions

**Comments to the Author**

Reviewer #1: All comments have been addressed

2. Is the manuscript technically sound, and do the data support the conclusions?

Reviewer #1: Yes

3. Has the statistical analysis been performed appropriately and rigorously?

Reviewer #1: Yes

4. Have the authors made all data underlying the findings in their manuscript fully available?

Reviewer #1: No

5. Is the manuscript presented in an intelligible fashion and written in standard English?

Reviewer #1: Yes

Reviewer #1: The authors have done remarkable work to improve the robustness of the analyses reported in the manuscript. I particularly appreciate the inclusion of an additional analysis to distinguish the mu rhythm from occipital alpha activity. Likewise, I value the additional analyses conducted in the beta band.

I only have a few minor points:

• Is this a typo? Line 381: “Each orage dot”

• Results of machine learning: besides accuracy, it might be interesting to report other performance metrics.

• Line 710: I would rephrase this sentence without explicitly mentioning the statistical test: “BANOVA favored null hypotheses as for main effects of the task and imitative experience as …”

• Conclusions (Lines 737–741): I would rephrase the final sentence in a more cautious manner, also considering that, when discussing autism, different levels of severity exist. Moreover, the exact neurobiological mechanisms underlying this condition are not yet known.

• Line 744: if the acknowledgments section has been removed, the corresponding heading should also be deleted.

**Do you want your identity to be public for this peer review?** For information about this choice, including consent withdrawal, please see our Privacy Policy

Reviewer #1: No

---

## [Author Response · Author response to Decision Letter 2]

17 Oct 2025

Dear Reviewer1,

We are mostly grateful for additional suggestions that will further improve our manuscript. We seriously took your suggestions into consideration, and reflected them on the revised manuscript. We hope the revised manuscript successfully addressed your concerns.

Comment 1

Is this a typo? Line 381: “Each orage dot”

Response

Appreciating the reviewer’s comment, we fixed the typo on pp.17.

Comment 2

Results of machine learning: besides accuracy, it might be interesting to report other performance metrics.

Response

Following the plausible suggestion, we added precision, recall, and F1-score of the target and randomized machines on pp.24-25. We also included the results of permutation tests and their p-values.

Comment 3

Line 710: I would rephrase this sentence without explicitly mentioning the statistical test: “BANOVA favored null hypotheses as for main effects of the task and imitative experience as …”

Response

We followed the reviewer's comment and rephrased the sentence as below.

“While beta suppression in the central regions may correspond to excitation of the MNS, the suppression was not modulated by the task type, imitative experience, and their interaction.” (pp.32).

Comment 4

Conclusions (Lines 737–741): I would rephrase the final sentence in a more cautious manner, also considering that, when discussing autism, different levels of severity exist. Moreover, the exact neurobiological mechanisms underlying this condition are not yet known.

Response

As the reviewer suggested, autism is characterized by multidimensional variation, and its neurobiological underlying is less known. Because the present study was not directly related to autism, we decided to delete the sentences addressing the disorder in conclusion (pp.33)

Comment 5

Line 744: if the acknowledgments section has been removed, the corresponding heading should also be deleted.

Response

Following the reviewer’s comment, we removed the heading of acknowledgement on pp.34.

---

## [Editor Report · Decision Letter 2]

20 Oct 2025

Effects of Intention Understanding and Brief Imitative Experience on the Mirror Neuron System: An EEG Study using Japanese Sign Language

PONE-D-25-05068R2

Dear Dr. Minamoto,

We’re pleased to inform you that your manuscript has been judged scientifically suitable for publication and will be formally accepted for publication once it meets all outstanding technical requirements.

Kind regards,

Luigi Cattaneo, MD, PhD

Academic Editor

PLOS ONE
---

## [Editor Report · Acceptance letter]

PONE-D-25-05068R2

PLOS One

Dear Dr. Minamoto,

I'm pleased to inform you that your manuscript has been deemed suitable for publication in PLOS One. Congratulations! Your manuscript is now being handed over to our production team.

Kind regards,

on behalf of

Dr. Luigi Cattaneo

Academic Editor

PLOS One